# Enhanced Auditory Steady-State Response Using an Optimized Chirp Stimulus-Evoked Paradigm

**DOI:** 10.3390/s19030748

**Published:** 2019-02-12

**Authors:** Xiaoya Liu, Shuang Liu, Dongyue Guo, Yue Sheng, Yufeng Ke, Xingwei An, Feng He, Dong Ming

**Affiliations:** 1Academy of Medical Engineering and Translational Medicine, Tianjin University, Tianjin 300072, China; 18222932286@163.com (X.L.); shuangliu@tju.edu.cn (S.L.); dongyue_guo@tju.edu.cn (D.G.); clarenceke@tju.edu.cn (Y.K.); anxingwei@tju.edu.cn (X.A.); 2College of Precision Instruments & Optoelectronics Engineering, Tianjin University, Tianjin 300072, China; xiandaiguanli11@163.com (Y.S.); heaven@tju.edu.cn (F.H.)

**Keywords:** electroencephalogram, gamma band, chirp ASSRs, click ASSRs, event-related spectral perturbation, signal-to-noise ratio

## Abstract

**Objectives**: It has been reported recently that gamma measures of the electroencephalogram (EEG) might provide information about the candidate biomarker of mental diseases like schizophrenia, Alzheimer’s disease, affective disorder and so on, but as we know it is a difficult issue to induce visual and tactile evoked responses at high frequencies. Although a high-frequency response evoked by auditory senses is achievable, the quality of the recording response is not ideal, such as relatively low signal-to-noise ratio (SNR). Recently, auditory steady-state responses (ASSRs) play an essential role in the field of basic auditory studies and clinical uses. However, how to improve the quality of ASSRs is still a challenge which researchers have been working on. This study aims at designing a more comfortable and suitable evoked paradigm and then enhancing the quality of the ASSRs in healthy subjects so as to further apply it in clinical practice. **Methods**: Chirp and click stimuli with 40 Hz and 60 Hz were employed to evoke the gamma-ASSR respectively, and the sound adjusted to 45 dB sound pressure level (SPL). Twenty healthy subjects with normal-hearing participated, and 64-channel EEGs were simultaneously recorded during the experiment. Event-related spectral perturbation (ERSP) and SNR of the ASSRs were measured and analyzed to verify the feasibility and adaptability of the proposed evoked paradigm. **Results**: The results showed that the evoked paradigm proposed in this study could enhance ASSRs with strong feasibility and adaptability. (1) ASSR waves in time domain indicated that 40 Hz stimuli could significantly induce larger peak-to-peak values of ASSRs compared to 60 Hz stimuli (*p* < 0.01**); ERSP showed that obvious ASSRs were obtained at each lead for both 40 Hz and 60 Hz, as well as the click and chirp stimuli. (2) The SNR of the ASSRs were –3.23 ± 1.68, –2.44 ± 2.90, –4.66 ± 2.09, and –3.53 ± 3.49 respectively for 40 Hz click, 40 Hz chirp, 60 Hz click and 60 Hz chirp, indicating the chirp stimuli could induce significantly better ASSR than the click, and 40 Hz ASSRs had the higher SNR than 60 Hz (*p* < 0.01**). **Limitation**: In this study, sample size was small and the age span was not large enough. **Conclusions**: This study verified the feasibility and adaptability of the proposed evoked paradigm to improve the quality of the gamma-ASSR, which is significant in clinical application. The results suggested that 40 Hz ASSR evoked by chirp stimuli had the best performance and was expected to be used in clinical practice, especially in the field of mental diseases such as schizophrenia, Alzheimer’s disease, and affective disorder.

## 1. Introduction

### 1.1. Background

In recent years, high-frequency brain activities at 30–80 Hz (gamma band) and above have become the focus of a growing body of work in electroencephalogram (EEG) research [1,2]. Researchers have demonstrated that high-frequency information of EEG might have a great potential to detect biomarkers of mental diseases (the diagnosis of neuropsychiatric disorders such as schizophrenia, Alzheimer’s disease (AD), major depressive disorder, affective disorder and so on) [2,3]. However, gamma-band EEG during rest is weak and difficult to extract. At present, visual, auditory and tactile stimuli are mostly used to generate evoked EEG responses, but there is a bottleneck in inducing high-frequency evoked EEG responses through visual and tactile stimulation. For example, subjects would feel fatigue under the high-frequency visual stimulation, and it could even induce epilepsy [4]. Therefore, auditory stimulus becomes a better choice to explore the characteristics of high-frequency EEG [5,6]. 

Cerebral evoked responses can be classified broadly into transient and steady-state responses depending on the time course of stimulation. Response evoked by auditory stimulus at inter-stimulus intervals long enough to allow intervening activity to subside, is called auditory transient-evoked responses (AEPs) [7,8]. When the inter-stimulus interval is short such that successive responses begin to overlap, a complex superimposed response is generated, which is referred to as an auditory steady-state response (ASSR) [9]. The ASSR is one of the most widely investigated responses with respect to gamma band neural oscillations [10,11], which could obtain better results compared to auditory brain stem response (ABR). Previous ASSR studies [12,13] have reported that patients with schizophrenia and depressive disorder show an altered ASSR for 40 Hz click stimuli. Therefore, ASSRs have become one of the most popular research topics in recent years.

### 1.2. The Auditory Steady-State Responses (ASSRs)

The ASSRs are electrophysiological responses which evoked by a periodically repeated auditory stimulus. During the stimulation, the responses are stable over time. While periodic auditory stimulation entrains the EEG to produce ASSRs that synchronize to the phase and frequency of the external stimulus [14]. The ASSRs could, therefore, probe frequency response characteristics sensory neural circuits with respect to phase synchronization and response magnitude. The ASSRs could reflect the activity of neurons in the brainstem, thalamocortical projections, and auditory cortex [15]. Currently, ASSRs are typically evoked by amplitude-modulated (AM)/frequency-modulated (FM) tonal stimuli or by repeated short tone bursts or clicks. Gamma-band ASSRs evoked by repeated short bursts or clicks have lower SNR due to high frequency noise interference [16]. In order to improve the SNR of ASSRs, this study introduced chirps, and explored whether the SNR of ASSRs evoked by chirp stimuli were enhanced compared with click stimuli.

### 1.3. The Chirp Stimulus

Recently, some optimized chirp stimuli have been designed to produce simultaneous displacement maxima along the basilar membrane by compensating for frequency-dependent traveling time [17,18,19,20]. The chirp, more specifically the upward chirp, is a type of stimulus designed to compensate for the cochlear delay of the traveling wave along the cochlear partition, as it travels from the base of the cochlea to the apex, i.e., from the highest to the lowest frequency responding area. The individual areas along the cochlear partition and the corresponding hair cells and nerve fibers of the auditory nerve will not be stimulated at the same time. This temporal dispersion can be counteracted by delaying the higher frequencies relative to the lower frequencies of the stimulus [19,21]. Such a scheme has to be based on an appropriate model of the traveling wave delay. A chirp stimulus attempts to compensate for the dispersion by aligning the arrival time of each frequency component in the stimulus to its place of maximum excitation along the basilar membrane. Such compensation will make the stimulus more efficient by achieving higher temporal synchronization between the evoked activities from the different neural elements that contribute to the formation of ASSRs [22,23]. The chirps have broader frequency band (100 Hz–10 kHz) [24] which could result in larger amplitude to improve the efficiency of frequency-specific ASSRs. 

### 1.4. Literature Review

The following references were listed to show the comparison of click- and chirp-stimuli and the research conclusions of ASSRs. Lütkenhöner constructed a chirp stimulus using the parameters for the cochlea traveling delay, and revealed that the chirp resulted in higher neural synchronization, larger ABR amplitude, and better response detection than the corresponding click [22]. Dau developed a “flat-spectrum chirp” [25] using the inverse delay-line characteristics of the cochlear partition. ABRs were recorded in 10 normal-hearing subjects and it was found that for most of the applied stimulus levels the flat-spectrum chirp generated ABRs with higher wave-V amplitude than the corresponding click stimulus. It was further concluded that in contrast to the click, the rising frequency chirp enables the inclusion of activity from lower cochlear frequency regions. Most studies have reached a consistent conclusion that chirps give shorter detection time and higher SNR than clicks [18,22]. At present, ASSRs were analyzed in response to auditory stimuli delivered over a range of frequencies (low, 30–40 Hz and high gamma, 60–80 Hz) in most auditory research, suggesting that, in humans, the ASSRs are largest for clicks or modulated tones presented at 40 Hz and 80 Hz [26,27,28]. In conclusion, there is much research on the comparison of ABR elicited by chirp- and click-stimuli, but little research on the comparison of ASSRs elicited by chirp- and click-stimuli, especially on the field of mental diseases. Based on previous research results, this study boldly hypothesizes that ASSRs evoked by chirp are more responsive and have a higher SNR than that evoked by click. In addition, 40 Hz was selected as the stimulation frequency in this study. Since the subjects in the preliminary experiment subjectively responded that the sound of 80 Hz was unbearable, the sound of 60 Hz in the high frequency band, which was more comfortable, was selected to replace 80 Hz.

### 1.5. The Present Study

The ASSRs comparison between chirp and click is intriguing, since it might have multiple implications regarding everyday practice. However, the literature is currently lacking such comparisons. So we expected that the improvement in response amplitude observed for the ABR by using a chirp instead of a click also applies to the ASSRs using high rates of stimulation [29]. Therefore, this study is not only to verify the feasibility and adaptability of the evoked paradigm, but also to analyze the difference between the traditional click and chirp evoked electrophysiological ASSRs by testing in a group of 20 healthy adults with normal hearing to explore whether the optimized chirp stimulus paradigm could induce stronger ASSRs. 

## 2. Materials and Methods

### 2.1. Subject

Twenty healthy students (10 females and 10 males; age range: 22–29 years; mean age: 23.85 years; SD = 1.57 years) of Tianjin University were recruited to participate in this study. All subjects were right-handed [30], with no history of hearing problems and their audiometric thresholds were 15 dB hearing level (HL) or better.

Exclusion criteria [31] for all participants included a history of neurological or cardiovascular disease, failure to complete a grade-school education, clinically documented hearing loss, head injury that resulted in loss of consciousness, current substance or alcohol dependence and electroconvulsive therapy. Participants were paid for their participation and the protocol was approved by the ethical committee of Tianjin University. All participants received detailed information about the study protocol and gave written and oral informed consent before participation.

### 2.2. Stimuli

Dau et al [32] described several forms of chirp stimuli and these chirp designs have been proposed and tested. This study only introduces three different chirp stimuli described by Dau, which are commonly used. One of the chirps, the O-chirp, was based on estimates of human basilar membrane (BM) group delays derived from stimulus-frequency otoacoustic emissions (SFOAE) [33,34]. Another chirp, the M-chirp, was based on cochlear delay values. The third chirp, the A-chirp, was based on cochlear delay values derived from the latencies of frequency-specific tone-burst ABRs in normal-hearing subjects, which resulted in cochlear delays that changed significantly with stimulus level [21,26,35]. The results of their study indicated that there was no substantial benefit of using the M-chirp compared with the O-chirp for ABR, and the design of the A-chirp requires knowledge of an individual’s hearing thresholds to determine the time course of the stimulus. The evoked paradigm is expected to be applied to as much as possible to a larger population, such as patients with mental diseases. Therefore, the O-chirp was used in present study, which is easy to construct and less affected by the hearing thresholds, to ensure the auditory comfort of most people and the high-quality signal obtained at the same time.

The O-chirp stimulus was constructed in MATLAB using the following equations:(1)$$s_{O}\left(t\right)=A_{O}\left(t\right)\sin\left[\emptyset_{O}\left(t\right)-\emptyset_{0}\right],$$ whereby $\emptyset_{0}$ determines the starting phase of the chirp, the phase $\emptyset_{O}\left(t\right)$ was chosen as:(2)$$\emptyset_{O}\left(t\right)=2\pi c^{2}\left(\frac{1}{t_{0}-t}-\frac{1}{t_{0}}\right),$$ and the amplitude factor $A_{O}\left(t\right)$ was given by:(3)$$A_{O}\left(t\right)=\sqrt{\frac{df_{O}\left(t\right)}{dt}}=\sqrt{\frac{2c^{2}}{{\left(t_{O}-t\right)}^{3}}},$$
(4)$$t_{O}=c{\left(50Hz/\left[Hz\right]\right)}^{\alpha },$$

In order to produce a stimulus with a flat magnitude spectrum. With the constants c=1.5 s, and α=−0.5, the $f_{O}\left(t\right)$ was chosen as:(5)$$f_{O}\left(t\right)={\left(\frac{c}{t_{0}-t}\right)}^{2}$$

Since the stimulus $s_{O}\left(t\right)$ is based on OAE data, it is referred to as the “O-chirp”. The schematic representation of the “O-chirp” stimuli can be seen in Figure 1a.

### 2.3. Design of the Evoked Paradigm

The duration of the click stimulus was 1-ms [6] and can be seen in Figure 1b. For chirp and click stimulus, the carrier was 44,100 Hz and the modulating envelope was 40 Hz/60 Hz. Sounds were presented binaurally through headphones AKG-k72 (AKG, Inc., Vienna, Austria) with 45 dB sound pressure level (SPL). The SPL of the stimulus calculated by deconvolving (using LABVIEW 3.1, made in National Instruments, Austin, TX, USA) the sound pressure waveform measured at the end of the probe tube and the probe-tube impulse response (the impulse response was calculated by a reverse Fourier transform of the measured probe-tube frequency-transfer function). Each trial of clicks and chirps lasted 3 s with the inter-stimulus interval set 1.5 s. Each trail was presented 56 times resulting in ∼4 min of stimulation in each block. The duration of each trial was extended from the previous milliseconds to seconds in order to induce a more stable ASSRs in the evoked paradigm. Chirp and click stimuli each have 3 blocks, a total of 6 blocks in the paradigm. 40 and 60 Hz stimulus trails randomly and evenly presented in each block, and the order of blocks of clicks and chirps was randomized across participants. The specific process can be seen in Figure 2.

During the evoked paradigm, the background noise intensity of the experimental environment was required to be less than 15 dB, and participants were seated comfortably with eyes open while listening to trains of clicks and chirps. Subjects were not required to perform any tasks. The total time of the experiment was 24 min, except for the rest time required by the subjects between each block. 

Apart from completing the auditory experiment, for subjects, they are also required to subjectively evaluate the tolerability of the evoked paradigm, with the score ranging from 0 to 4. For details, please refer to Figure 3. The lower the score, the lower the tolerance.

### 2.4. Data Acquisition and Processing

#### 2.4.1. Data Acquisition

The SynAmps2 electroencephalography system produced by the United States Neuroscan was used in this study. In the experiment, 64-channel EEGs was recorded continuously from Ag/AgCl-electrodes according to the international 10/20 system of electrode placement, and all channels were referenced to left mastoid (‘M1’) with the ground electrode located at ‘AFz’. The signals were digitized at 1000 Hz and stored in a PC for offline analysis. Throughout the EEG recording, the impedances of the electrodes were maintained below 5 kΩ.

#### 2.4.2. Data Processing

**Offline preprocessing:** all channels were re-referenced to bilateral mastoids (‘M1’ and ‘M2’), down-sampled to 500 Hz and used a band-pass filter range of 0.5 to 80 Hz to reduce DC interference and high-frequency noise. Besides the two references electrodes, ‘CB1’ and ‘CB2’ channels, the remaining off-line 60-lead EEG data processing was performed in MATLAB.

**Event-related spectral perturbation (ERSP):** the ERSP image, a time/frequency measures and equivalent 2-D images, reveal the frequencies and latencies when mean changes in power occur from some mean power baseline, time-locked to a class of experimental events. Subtracting mean baseline power measures depends on how strongly mean, event-related power at different frequencies either increased or decreased relative to the baseline spectrum [36,37,38]. Generally, the short-time Fourier transform (STFT) could be used to computer ERSP. In practice, the procedure for computing STFT is to divide a longer time signal into shorter segments of equal length and then compute the discrete Fourier transform (DFT) separately on each shorter segment. In contrast to the DFT, the STFT usually plots the changing spectra as a function of time. The STFT [39] is said to provides the best compromise between spectral and temporal resolution and thus is the most appropriate approach for the analysis of EEG data. In this study, the epochs to click/chirp stimuli were extracted using a time window of 3.2 s (0.2 s pre-stimulus and 3s post-stimulus) and baseline corrected by demeaning the EEG activity within the pre-stimulus interval. Then ERSP was obtained through the overlapping average of multiple trails in order to further determine whether the stimulation of 40 Hz and 60 Hz could induce the same frequency ASSRs and the changes of ASSRs over time.

**Signal-to-noise ratio (SNR):** SNR or S/N is defined as the ratio of the power of a signal (meaningful information) to the power of background noise (unwanted signal) [40,41]:(5)$$SNR=\frac{P_{signal}}{P_{noise}}$$
where *P* is average power. Similarly, $P_{noise}$ is average power of background noise and $P_{signal}$ is average power of signal. For the 40 Hz stimuli, 0–30 Hz and 50–80 Hz were selected as the background noise bands and 30–50 Hz was selected as the signal band in this study. For the 60 Hz stimuli, 0–50 Hz and 70–80 Hz were selected as the background noise bands and 30–50 Hz was selected as the signal band in this study.

## 3. Results

### 3.1. Statistical Analyses

All statistical analyses were carried out using the Statistical package for the social sciences for Windows (version 19.0; IBM Corporation, Armonk, NY, USA). Demographic characteristics such as age and years of education in the healthy subjects of different genders were compared by one-way analysis of variance (ANOVA) [42]. Significant difference was set at *p* value of less than 0.05. 

Demographic characteristics and behavioral data are shown in Table 1. There were no significant differences between male and female healthy groups in age and education. Both male and female healthy subjects had completed all the blocks of the auditory experiment, and by rating the scores of the subjects on the tolerance of stimulation, we could see that all subjects scored totally tolerable, indicating that the evoked paradigm is acceptable for most people no matter male or female.

### 3.2. The ASSR Waves in Time Domain

Since the auditory pathway is related to the central parietal region and temporal lobe of the brain [43], this work selected the leads of the central parietal and temporal lobe regions (‘FT7’, ’FC5’, ’FC3’, ’FC1’, ’FCZ’, ’FC2’, ’FC4’, ’FC6’, ’FT8’, ’T7’, ’C5’, ’C3’, ’C1’, ’CZ’, ’C2’, ’C4’, ’C6’, ’T8’, ’TP7’, ’CP5’, ’CP3’, ’CP1’, ’CPZ’, ’CP2’, ’CP4’, ’CP6’, ’TP8’) for time-domain analysis. The mean amplitudes and peak-to-peak values of ASSRs from 20 healthy subjects were shown in Figure 4a, b respectively. Narrow band-pass (40–80 Hz band-pass was conducted on 40 Hz stimuli and 50—80 Hz band-pass on 60 Hz stimuli) on the time sweeps and Paired t-test were used to assess the difference of the time-domain measures. From Figure 4a, it is obvious that chirp stimuli could increase the amplitude of ASSRs compared to click stimuli. Meanwhile, the amplitude of 40 Hz ASSRs was larger than that of 60 Hz ASSRs. As can be seen from Figure 4b, the peak-to-peak value of ASSRs induced by 40 Hz chirp was increased 73.3% than that by the 40 Hz click, while there was no significant difference between the peak-to-peak value induced by 40 Hz chirp and that by the 40 Hz click (*p* = 0.07 > 0.01), and the same result could be seen between the 60 Hz stimuli, namely, there were no significant difference between the chirp and click stimuli at the corresponding frequency. However, the peak-to-peak value of 40 Hz ASSRs were greater than 60 Hz ASSRs significantly (*p* < 0.01) for both chirp and click stimuli.

### 3.3. Event-Related Spectral Perturbation (ERSP) Analysis

To determine whether 40 Hz and 60 Hz stimuli could successfully induce ASSRs at the corresponding frequency, ERSP was computed for all leads. We found that each stimulation could evoke corresponding frequency ERSP for each lead significantly. Figure 5 depicted the mean ERSP averaged across all leads from 20 healthy subjects. It could be found that, whether click or chirp, 40 Hz and 60 Hz stimuli could induce apparent ERSP at the same frequency, and ERSP from chirp stimuli were more obvious than that from click. However, it seems to there was no obvious difference between the ERSP evoked by 40 Hz chirp and 60 Hz chirp. 

As we could see, both click and chirp at 40 Hz and at 60 Hz could induce the same frequency ASSRs apparently, indicating that the proposed auditory evoked paradigm could induce and enhance ASSRs with significant effect successfully. Furthermore, chirp stimuli could induce a larger ASSRs than click stimuli at both 40 Hz and 60 Hz frequencies. However, it does not appear to show a significant difference between the ASSRs evoked by chirps at 40 Hz and at 60 Hz. Therefore, we might hypothesize that 60 Hz stimulation might also induce better ASSRs and can be explored further.

### 3.4. Signal-to-Noise Ratio (SNR) Analysis

The SNRs of ASSRs were calculated and averaged across 20 subjects, shown in Figure 6a. It could be found that the chirp ASSR had a higher SNR than the click for both 40 Hz and 60 Hz, especially in the central parietal and temporal lobe regions. It is interesting that 40 Hz click could induce the ASSR with relatively higher SNR than the 60 Hz chirp in the central parietal of the brain. 

To depict more clearly, mean SNRs were calculated across all leads, shown in Figure 6b. It can be seen that mean SNR of ASSRs evoked by 40 Hz chirp (–2.44 ± 2.90) was higher than that evoked by a 40 Hz click (–3.23 ± 1.68), and the mean SNR of ASSRs evoked by 60 Hz chirp (–3.53 ± 3.49) was higher than that evoked by a 60 Hz click (–4.66 ± 2.09). Meanwhile, the mean SNRs of ASSRs evoked by 40 Hz stimuli were higher than 60 Hz stimuli for both click and chirp.

To have a closer look at the SNR of the ASSRs, Figure 7a, b showed the SNR histogram at each lead. As can be seen in Figure 7a, the SNR of ASSRs under 40Hz chirp appeared significantly higher than that under 40Hz click at the leads of ‘FP1’, ‘AF3’, ‘F5’, ‘F3’, ‘F1’, ‘FZ’, ‘F2’, ‘F8’, ‘FT7’, ’FC5’, ’FC3’, ’FC1’, ’FCZ’, ’FC2’, ’FC4’, ’C5’, ’C3’, ’C1’, ’CZ’, ’C2’, ’C4’, ’C6’, ’T8’, ’TP7’, ’CP5’, ’CP3’, ’CP1’, ’CPZ’, ’CP2’, ’CP4’, ‘P7’, ‘PZ’, mostly located at the central parietal region and the temporal lobes of the brain, which are related to the auditory pathway. The results above indicated that the proposed auditory evoked paradigm could induce and enhance ASSRs with a higher SNR. Figure 7b showed the mean SNR histogram of 40 Hz chirp and 60 Hz chirp, and it could be found that the SNR of 40 Hz chirp ASSR was higher than that of 60 Hz chirp at nearly all leads, indicating that the 40 Hz chirp is the best choice to evoke the gamma-ASSRs.

## 4. Discussion

To our knowledge, this is the first ASSRs study to compare chirps and clicks at 40 Hz and 60 Hz. The behavioral analysis of the tolerance score showed that the evoked paradigm can be accepted by most people, namely, it has high feasibility and adaptability. In the preliminary experiment, chirp and click stimuli of 20 Hz and 80 Hz [44,45] were also selected into this study. However, healthy subjects have a low tolerance score for the sound of 80 Hz stimulus, and the ERSP of ASSRs evoked by 20 Hz stimulus is not as significant as the ERSP of ASSRs evoked by 40 Hz and 60 Hz, only 40 Hz and 60 Hz sounds were selected in the final formal experiment. 

In addition, the results of ASSR waves in the time domain showed that the 40 Hz stimuli could induce more significant ASSRs than the 60 Hz stimuli, indicating that the 40 Hz ASSR induced by the evoked paradigm, is expected to be a potential biomarker, consistent with previous reports [6,27,28,46]. Furthermore, from the results of ERSP, it can be seen that both click and chirp at 40 Hz and at 60 Hz could induce the same frequency ASSRs apparently at each lead, and the ERSPs induced by chirp were seems enhanced compared to that evoked by click both at 40 Hz and 60 Hz. The conclusion in ERSP analysis, was consistent with the result of average peak-to-peak value of 40 Hz click and 40 Hz chirp in the time domain analysis. However, we drew a conclusion inconsistent with the time domain analysis at 60 Hz. As the ERSP result showed, ERSP evoked by the 60 Hz chirp was greater than that by the 60 Hz click, but there was nearly no difference in the average peak-to-peak values between the two sound types. Therefore, we speculated the phenomenon was related to the more intensive peak-to-peak value distribution under the 60 Hz chirp stimulus (see Figure 4b), which leads to the larger area under the response waveform. Moreover, from the results of SNR analysis, as shown in Figure 6a and Figure 7, we could find a phenomenon under the leads of ‘F5’, ‘F3’, ‘F1’, ‘FZ’, ‘F2’, ‘FT7’, ’FC5’, ’FC3’, ’FC1’, ’FCZ’, ’FC2’, ’FC4’, ’FC6’, ’FT8’, ’T7’, ’C5’, ’C3’, ’C1’, ’CZ’, ’C2’, ’C4’, ’C6’, ’T8’, ’TP7’, ’CP5’, ’CP3’, ’CP1’, ’CPZ’, ’CP2’, ’CP4’, ’CP6’, ’TP8’, the ASSRs induced by the evoked paradigm in proposed this work was more significant. This is consistent with numerous studies showing that the activity of auditory pathway is connected with the temporal lobe and the central parietal region of the brain [43,47,48,49,50]. Therefore, the analysis of ASSRs under these leads would be analyzed and verified in further studies. 

This study has certain limitations. First, our sample size was small. Second, the age range of subjects was relatively concentrated, and further experiments to expand the age range need to be carried out. Third, the analysis of phase synchronization of ASSRs needs further exploration. Fourth, we could find that stimulation at 40 Hz also induces some frequency multiples, so the frequency multiples will be taken into account. This study also provides technical support for the clinical research; for example, it can be applied to the study of physiological and pathological mechanisms of patients with mental diseases, like schizophrenia, AD, major depressive disorder, affective disorder and so on.

The physiological and pathological mechanism of mental diseases (such as schizophrenia and major depressive disorder) is not clear, and there are no objective and systematic biomarkers as a diagnostic standard, leading to suboptimal treatment and poor outcomes. Thus, discriminating mental diseases at earlier stages of illness could help to facilitate efficient and specific treatment. Recently, the ASSR elicited by gamma band neural oscillations has received considerable interest as a biomarker of psychiatric disorders in much research as mentioned above. Therefore, in the present study, we are working in investigating low (ASSR for 40 Hz click/chirp trains), and high gamma [2,51,52] (ASSRs for 60 Hz click/chirp trains) ASSRs in healthy subjects and patients with mental diseases, such as schizophrenia, AD, and affective disorder and so on. We sought to evaluate whether ASSRs in mental diseases are different from that in healthy individuals, in order to explore and find the electrophysiological markers of healthy subjects and some common mental disorders mentioned above.

## 5. Conclusions

In this study, we verified the feasibility and adaptability of the evoked paradigm, and confirmed that the ASSRs could be enhanced in the evoked paradigm. In addition, the chirp’s high SNR was verified. Furthermore, ASSR evoked by 40 Hz chirp stimulus is more significant than that evoked by other stimuli, suggesting that the 40 Hz ASSR induced by chirp in the evoked paradigm is expected to be a potential biomarker to analysis gamma band EEG. However, further studies are needed for the small size of the subject population. In the future, we will carry out a clinical study to observe patients with mental diseases like schizophrenia and depression disorder by using the evoked paradigm in this study, so as to explore the physiological and pathological mechanism of mental diseases patients and help them recover as soon as possible.

## Figures and Tables

**Figure 1 sensors-19-00748-f001:**
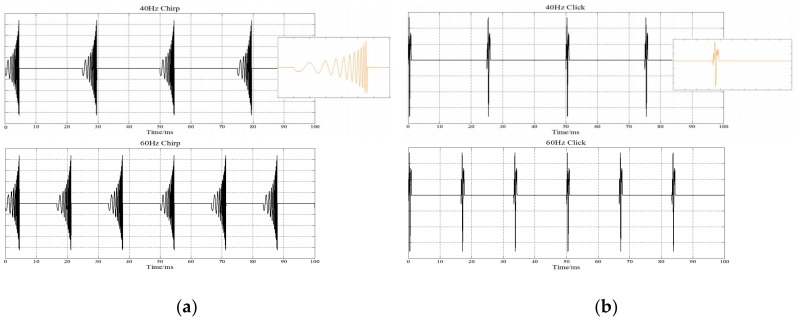
The schematic representation of the click and chirp stimuli: (**a**) the waveform of the chirp stimuli; (**b**) the waveform of the click stimuli. 40 Hz stimuli waveform at the top and 60 Hz stimuli waveform at the bottom in each subgraph.

**Figure 2 sensors-19-00748-f002:**
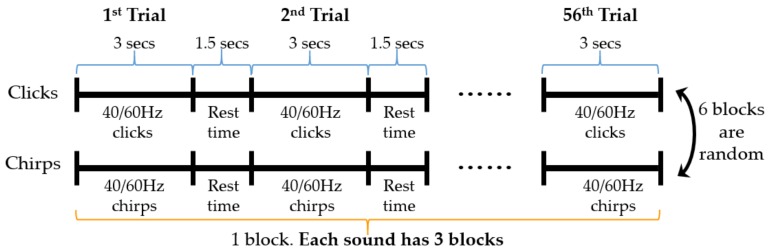
The flow chart of auditory evoked paradigm in this study: each frequency is represented 28 times in each block.

**Figure 3 sensors-19-00748-f003:**
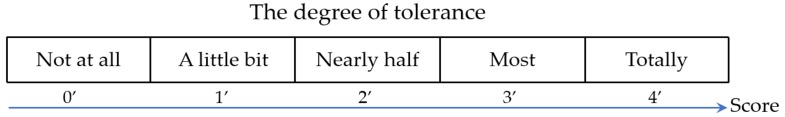
The scoring criteria of the degree of tolerance for click stimuli and chirp stimuli in the evoked paradigm.

**Figure 4 sensors-19-00748-f004:**
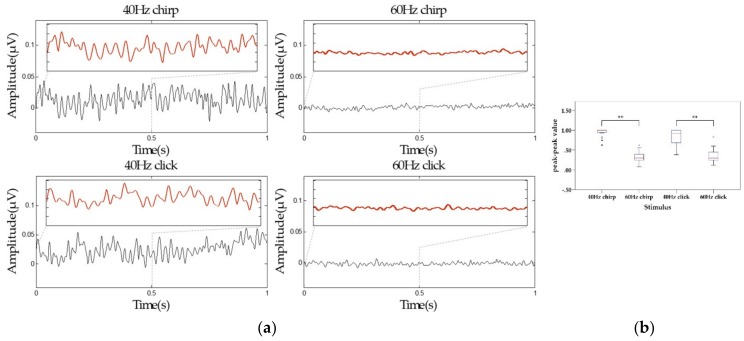
Mean time-domain waveforms of auditory steady-state responses (ASSRs) for 20 healthy subjects under selected leads: (**a**) Mean ASSR amplitude: the black line represents ASSR amplitude within 1s after stimulus, and the red line represents ASSR amplitude within 0.5 s after stimulus; (**b**) The boxplot of mean ASSR peak-peak value: the red line represents the median; ■ and ° represent outliers; ** represents *p* ≤ 0.01.

**Figure 5 sensors-19-00748-f005:**
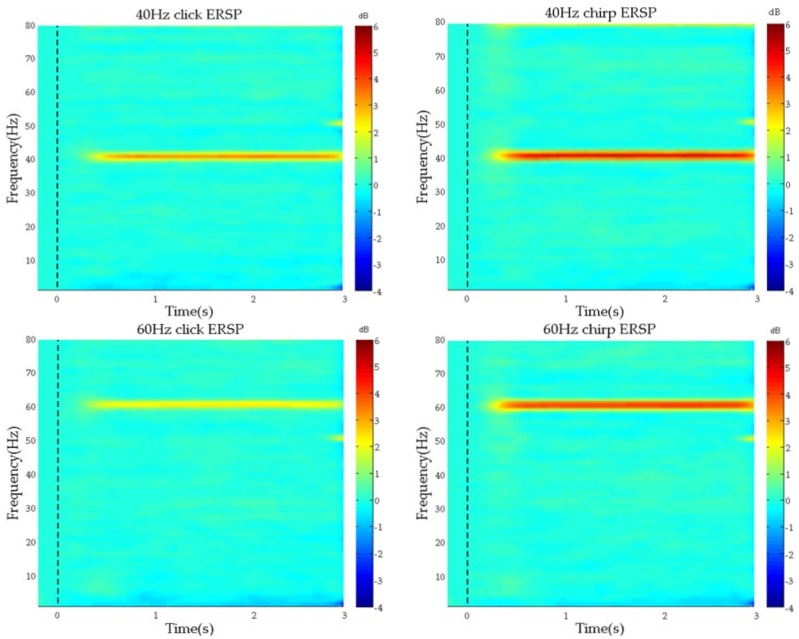
The event-related spectral perturbation (ERSP) averaged across all leads from 20 subjects: 40 Hz ERSP at the top and 60 Hz ERSP at the bottom. The left column represents the click stimuli and the right represents chirp stimuli. The dotted black lines represent the stimulus moment, namely the 0 time. The larger the number on the color bar, the stronger the evoked ASSRs response.

**Figure 6 sensors-19-00748-f006:**
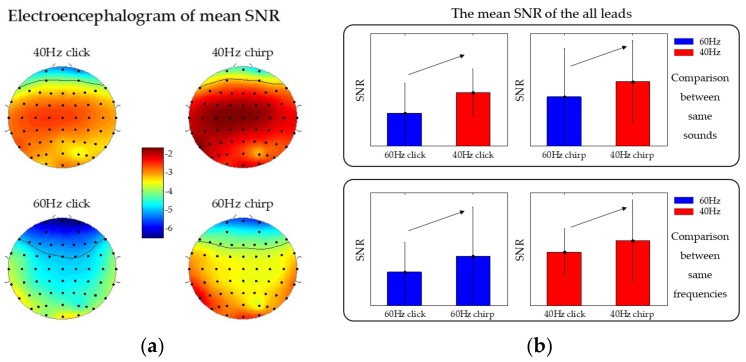
Mean signal-to-noise (SNR) electroencephalogram across 20 subjects: (**a**) 40 Hz electroencephalogram at the top and 60 Hz electroencephalogram at the bottom. The left column represents the click stimuli and the right represents chirp stimuli; (**b**) histogram of the mean SNR of ASSR under the whole leads.

**Figure 7 sensors-19-00748-f007:**
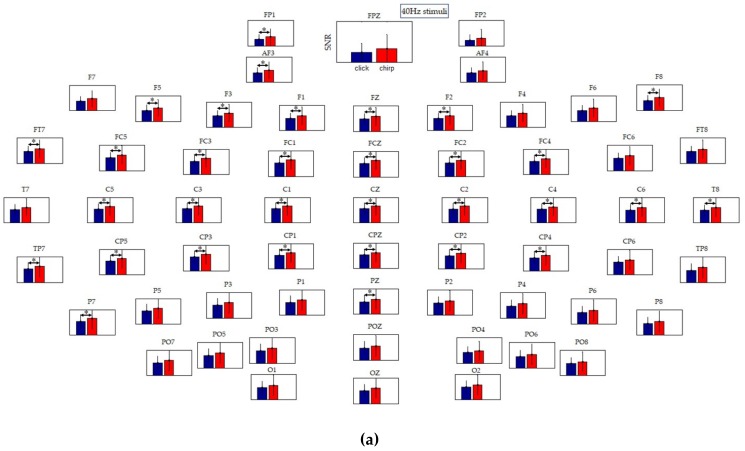
Mean SNR histogram at each lead across 20 subjects: (**a**) the mean SNR histogram of 40 Hz chirp stimulus and 40 Hz click stimulus; (**b**) the mean SNR histogram of 40 Hz chirp stimulus and 60 Hz chirp stimulus; * represents *P* ≤ 0.05, ** represents *P* ≤ 0.01.

**Table 1 sensors-19-00748-t001:** One-way analysis of variance (ANOVA) of male and female healthy subjects.

	MHS (N=10)	FMHS (N = 10)	F	Significance.
**Age (years)**	23.40 ± 1.174	24.30 ± 1.829	1.715	0.207
**Education (years)**	17.50 ± 0.527	18.50 ± 1.581	3.600	0.074
**Number of completed blocks**	60	60	-	-
**Tolerance score (total score)**	40	40	-	-

MHS, Male Healthy Subjects; FMHS, Female Healthy Subjects.

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
