# Peer review of "Enhanced Auditory Steady-State Response Using an Optimized Chirp Stimulus-Evoked Paradigm"

_sensors, 2019, doi:10.3390/s19030748_

Round 1
Reviewer 1 Report
The article is interesting and presents potential for future work. However, there are some observations that it would be good to clarify:
- In eq. 2 ad 3: what are the values of c and to?
- In Fig. 5 it is not clear if the ERSP is calculated on a electrode, since it is very similar to a spectrogram. In this case it would be convenient to indicate to which electrode it corresponds.
- I'm not sure if Fig. 6 is useful, since you can not clearly see the ERsP graphs on each electrode.
- What are the electrodes used to calculate the histograms in Fig. 7.b?
Author Response
Dear reviewer
Thank you very much for your patient revision and precious advices. We have revised the article carefully following the suggestions. And we hope you will be satisfied with the new version and our reply.
[1] In eq. 2 ad 3: what are the values of c and to?
Thank you very much for your advice. We are sorry about that the values were not clearly presented in the previous version. The constants c=0.15s and t0=0.0212, In the new version, we have given the value of c and the equation of t0 to make it clearer to readers.
[2] In Fig. 5 it is not clear if the ERSP is calculated on an electrode, since it is very similar to a spectrogram. In this case it would be convenient to indicate to which electrode it corresponds.
[3] I'm not sure if Fig. 6 is useful, since you cannot clearly see the ERSP graphs on each electrode.
Thank you very much for your advice. The Event-related spectral perturbation (ERSP) plots the changing spectra as a function of time. Fig.5 showed the ERSPs averaged across all the leads,and the ERSP of each lead was shown in Fig. 6 in the previous version, but Fig.5 could illustrate the results clearly, so we deleted Fig.6 and meanwhile added the related description in the new version.
[4] What are the electrodes used to calculate the histograms in Fig. 7.b?
Thank you very much for your question. The 60-leads were used to calculate the histograms in Fig. 7.b in this study. As shown in Fig. 6.b in the new version, we have added the related description.

Reviewer 2 Report
Enhanced auditory steady-state response using an optimized chirp stimulus evoked paradigm
Review: 01/22/2018
1. General comments
The manuscript presents a paradigm designed to optimize the auditory steady state recordings using chirps. The significance of the study is not very well supported by the cited literature. The authors should better identify the possible applications of the paradigm. Audiometry is mentioned, but the use of stimuli with broad spectrum make it not feasible. If there are still diseases or conditions (schizophrenia diagnosis, anesthesia monitoring , ..) that my benefit from the developed method; they should be mention.
The stimuli design (click) and calibration methods require revision (see specific comments). The population size and composition a-priori seem adequate to fulfill the study aims. The analysis methods in time domain can be improved by applying narrow band-pass on the time sweeps and by performing statistical test to assess the statistical significance of the results. The authors should discuss why the ERSP analysis is more convenient than the simple DFT approach.
2. Specific comments
2.1 Introduction:
Background: Lines 43-47.
The argument that the authors use to justify the use of high frequency auditory steady state responses (ASSRs) to overcome many difficulties of the analysis of high frequency components in the EEG is not very well supported by the two cited references. In reference [1] most of the applications study high frequency EEG components are obtained using low stimulation rates for which the concerns of [2] are not applicable.
I would agree that ASSRs have a great potential to detect biomarkers of mental diseases and conditions. The authors should expand the references to provides support the impact of the manuscript. I request the modification of the background section (and abstract)
2.2 Materials and Methods
Design of the Evoked paradigm: lines 153-157
What the authors define as a click is not what the literature does. The authors use a noise burst which is different from a click. A click is a stimulus of constant amplitude lasting very short time (100 or 200 microseconds).
The authors should explain in detail how the acoustical calibration is performed; the presented explanation mentions that the sounds were adjusted to 45 dB. There is not mention what kind of dB (dB nHL or SPL or ?).
2.3 Statistical analysis:
The authors should use statistical tests to assess the significance of the claims of the amplitude and SNR differences between chirp and click stimulation. The figures show the trends in the right direction but it shows high variability (standard deviation).
2.4 Discussion: The manuscript will improve if the authors better identify possible mental diseases where the paradigm can be applied.
Author Response
Dear reviewer
Thank you very much for your available advices and thought-provoking questions, which have helped us a lot. The new submission has been greatly revised. And the answers to your questions have been list in the following. We hope you will be satisfied with the new manuscript and our answers.
[1] The manuscript presents a paradigm designed to optimize the auditory steady state recordings using chirps. The significance of the study is not very well supported by the cited literature. The authors should better identify the possible applications of the paradigm. Audiometry is mentioned, but the use of stimuli with broad spectrum make it not feasible. If there are still diseases or conditions (schizophrenia diagnosis, anesthesia monitoring, …) that my benefit from the developed method; they should be mention.
Thank you very much for these thought-provoking questions which enrich our article a lot. According to your suggestion, we have added some literatures to support the significance of this study, and made corresponding supplements in the ‘Introduction’ and ‘Discussion’ sections to prove that the diagnosis of psychiatric diseases (e.g., schizophrenia, Alzheimer’s disease and affective disorder, etc.) would benefit from the experimental paradigm. Meanwhile, the ‘Introduction’ and ‘Discussion’ sections have been reorganized with a clearer structure in the new version.
[2] The stimuli design (click) and calibration methods require revision (see specific comments). The population size and composition a-priori seem adequate to fulfill the study aims. The analysis methods in time domain can be improved by applying narrow band-pass on the time sweeps and by performing statistical test to assess the statistical significance of the results. The authors should discuss why the ERSP analysis is more convenient than the simple DFT approach.
We have made some revision to the click stimuli design and calibration methods in the new version according your suggestions.
We have applied narrow band-pass on the time sweeps and performed statistical test to assess the statistical significance of the results in time domain, and the results were shown in section 2.1.
For ERSP analysis, we have presented a more detailed explanation in section 2.4.2 in the new version. Here, we integrate these parts to a concise and clear reply. The ERSP image, a time/frequency measures and equivalent 2-D images, reveal the frequencies and latencies when mean changes in power occur from some mean power baseline, time-locked to a class of experimental events. In this study, the short-time Fourier transform (STFT) was used to computer ERSP. In contrast to the DFT, the STFT usually plots the changing spectra as a function of time. And the STFT is reported to provide best compromise between spectral and temporal resolution. The ERSP diagram could present not only the energy change of a certain frequency at a particular moment, but also the change over time.
[3] Introduction: Background: Lines 43-47. The argument that the authors use to justify the use of high frequency auditory steady state responses (ASSRs) to overcome many difficulties of the analysis of high frequency components in the EEG is not very well supported by the two cited references. In reference [1] most of the applications study high frequency EEG components are obtained using low stimulation rates for which the concerns of [2] are not applicable.
I would agree that ASSRs have a great potential to detect biomarkers of mental diseases and conditions. The authors should expand the references to provides support the impact of the manuscript. I request the modification of the background section (and abstract).
Thank you very much for your nice advices, we have expanded the references to provides support the impact of the manuscript in the new version. And we have modified the ‘Background’ and ‘Abstract’ sections carefully.
[4] Materials and Methods Design of the Evoked paradigm: lines 153-157 What the authors define as a click is not what the literature does. The authors use a noise burst which is different from a click. A click is a stimulus of constant amplitude lasting very short time (100 or 200 microseconds).
The authors should explain in detail how the acoustical calibration is performed; the presented explanation mentions that the sounds were adjusted to 45dB. There is not mention what kind of dB (dB nHL or SPL or?).
Thank you very much for precious advices, we apologize for not defining the click stimuli clear. A click is defined a stimulus of constant amplitude lasting very short time (80 or 200 microseconds) in most literatures. We adjusted the 100 microseconds click stimuli as defined in most literatures to 1 millisecond in order to maintain the consistency with the clinical application (Isomura S, Onitsuka T, et al. Differentiation between major depressive disorder and bipolar disorder by auditory steady-state responses[J]. Journal of Affective Disorders, 2016, 190:800.), which used magnetoencephalography to explore the difference of ASSR between healthy subjects and depressed patients induced by click stimuli, but we didn’t make it clear. We are really sorry for that caused a lot of trouble to you, and we have re-define the click stimuli in the new version.
The acoustical calibration and the kind of the dB were not clearly presented in the previous version, but have been clarified in the new version.
[5] Statistical analysis: The authors should use statistical tests to assess the significance of the claims of the amplitude and SNR differences between chirp and click stimulation. The figures show the trends in the right direction but it shows high variability (standard deviation).
Thank you very much for your suggestions, and we have supplemented statistical tests to assess the significance of the amplitude of ASSR and SNR differences between chirp and click stimulation, and some statistically significant results were analyzed in the new version.
[6] Discussion: The manuscript will improve if the authors better identify possible mental diseases where the paradigm can be applied.
Thank you very much for this thought-provoking question which gave this part a great promotion. In ‘Discussion’ section, we have identified possible mental diseases which could benefit from the experimental paradigm in the new version.

Round 2
Reviewer 2 Report
The revision of the manuscript give it a focus in mental diseases. The references well support the selected application. The statistical analysis made the results stronger.